# Evaluation of Changes in Some Functional Properties of Human Mesenchymal Stromal Cells Induced by Low Doses of Ionizing Radiation

**DOI:** 10.3390/ijms24076346

**Published:** 2023-03-28

**Authors:** Daria Yu. Usupzhanova, Tatiana A. Astrelina, Irina V. Kobzeva, Yulia B. Suchkova, Vitaliy A. Brunchukov, Anna A. Rastorgueva, Victoria A. Nikitina, Alexander S. Samoilov

**Affiliations:** State Research Center-Burnasyan Federal Medical Biophysical Center of Federal Medical Biological Agency, 123098 Moscow, Russia

**Keywords:** mesenchymal stromal cells, immunology profile, secretory profile, proliferation, low dose, ionizing radiation

## Abstract

Each person is inevitably exposed to low doses of ionizing radiation (LDIR) throughout their life. The research results of LDIR effects are ambiguous and an accurate assessment of the risks associated with the influence of LDIR is an important task. Mesenchymal stromal cells (MSCs) are the regenerative reserve of an adult organism; because of this, they are a promising model for studying the effects of LDIR. The qualitative and quantitative changes in their characteristics can also be considered promising criteria for assessing the risks of LDIR exposure. The MSCs from human connective gingiva tissue (hG-MSCs) were irradiated at doses of 50, 100, 250, and 1000 mGy by the X-ray unit RUST-M1 (Russia). The cells were cultured continuously for 64 days after irradiation. During the study, we evaluated the secretory profile of hG-MSCs (IL-10, IDO, IL-6, IL-8, VEGF-A) using an ELISA test, the immunophenotype (CD45, CD34, CD90, CD105, CD73, HLA-DR, CD44) using flow cytometry, and the proliferative activity using the xCelligence RTCA cell analyzer at the chosen time points. The results of study have indicated the development of stimulating effects in the early stages of cultivation after irradiation using low doses of X-ray radiation. On the contrary, the effects of the low doses were comparable with the effects of medium doses of X-ray radiation in the long-term periods of cultivation after irradiation and have indicated the inhibition of the functional activity of MSCs.

## 1. Introduction

Throughout life, a person is inevitably exposed to low doses (from 10 to 100 mGy [1]) of ionizing radiation (LDIR). One can be exposed to background radiation, within the framework of medical diagnostics and treatment, from radioactive waste dumps, during professional activities and air travel, etc. [2]. The International Commission on Radiological Protection (ICRP) has designated critical values of LDIR for humans in the range from 20 to 50 mGy per year [1]. Taking into account the inevitably growing number of LDIR sources in the modern world and also the results of studies which demonstrate that some groups of people receive a cumulative radiation dose of 50 mGy/year [3], an accurate assessment of the risks associated with LDIR is an important public health task.

There are some contradictions in the published studies of the effects of LDIR. On the one hand, the results of some studies indicate the negative effect of LDIR. In particular, double-stranded breaks DNA accumulates in cells [4], and since each double-stranded break hypothetically has the ability to induce cellular transformation, this criterion is considered one of the most significant for assessing the dose effect. Based on this, the ICRP follows a threshold-free linear concept, according to which the effect is directly proportional to the received radiation dose [2]. Certainly, the described concept is applicable for predicting the effects of large doses of radiation; however, it is not always useful for predicting the effects of low doses because some researchers point to the stimulating effects of LDIR [5]. Mentions of such effects in recent studies suggest that in the range of LDIR, the effect may not be proportional to the received radiation dose; this is consistent with the threshold concept [6]. Thus, the research results of the development patterns of the effects are exerted by LDIR and the mechanisms underlying them are ambiguous, meaning this study seems actual.

Mesenchymal stromal cells are the regenerative reserve of an adult organism, and they act as a promising model for studying the effects of irradiation using low doses because it is not possible to evaluate the effects of LDIR on the human body as a whole. MSCs remain in the human body for a long period of time due to their ability to self-sustain. They can be exposed by several rounds of radiation and can accumulate and pass the changes that have occurred to the next generations of cells because MSCs have a differentiation potential. Finally, the changes that have occurred in MSCs—the regenerative reserve of the body—affect the human body as a whole. Thus, qualitative and quantitative changes in the characteristics of MSCs can be considered as criteria for assessing the risks of exposure by LDIR [2], and the MSCs in turn as a model for assessing the individual radiosensitivity of a person, in particular, people who work in the nuclear industry.

It is important to note that stem cells are characterized by a greater radiosensitivity in comparison with other types of cells in the body according to their regularity; the less a cell is differentiated, the more radiosensitive a cell is. However, it has been shown that the response of MSCs to radiation damage is different from the response of embryonic stem cells. The effect of radiation on embryonic stem cells stimulates them to enter apoptosis [7], whereas the stem cells of an adult exhibit a wide range of different options for protection against radiation damage [8]. They are able to compensate for the negative effects of radiation exposure by implementing reactions to the resulting damage, such as the enzymatic activity of ATM protein, the activation of cell cycle verification points, and the repair of double-stranded DNA breaks [9]. Additionally, the nuclear organization of the genetic material of MSCs, which is globally more open and favorable for gene expression, facilitates the process of stopping the cell cycle and DNA repair in damaged cells [10]. In general, it is important to take into account that the degree of radiosensitivity of stem cells is also determined by their age, the stage of the cell cycle [11], the source of production (niche), and the type of radiation source (X-ray, gamma-, beta-, etc.) when conducting research on MSCs models.

It is also important that the radiosensitivity of MSCs may depend on the gender of the donor. In studies on animal models, it has been repeatedly shown that female individuals are characterized by more pronounced radiosensitivity, demonstrated through changes in the proteome [12], gene expression [13], in particular, oncogenes and proto-oncogenes [14], brain function [15], cognitive abilities [16], etc. Similar patterns have also been observed for women and men in epidemiological studies [17,18].

One of the most important criteria of the MSCs functional properties are their secretory and surface antigens profiles (immunophenotype), as well as their proliferative activity. Human MSCs should express a minimum amount of antigens set on their surface: a >99% expression of CD73, CD90, and CD105, and also a <2% expression of CD34, CD45, and HLA-DR according to the criteria established by the International Society for Cell Therapy [19]. In addition, MSCs from various sources can exhibit the expression of some additional surface molecules, in particular, CD44. The MSCs surface antigens are cellular receptors involved in the implementation and regulation of various cellular functions because they are part of signaling cascades. The composition of the MSCs secretory profile is quite various and in large part depends on the source from which they were isolated: there are may be both qualitative and quantitative differences in it [20,21]. In particular, the secretory profile of MSCs include the following factors: IDO, IL-10, IL-6, IL-8, and VEGF [22]. Factors are released by MSCs into the environment which provides their interaction with other cell types and also influences the development of pro- and anti-inflammatory reactions. The MSCs proliferative activity in vitro reflects their ability to self-sustain and reproduce. The presented study is devoted to the research of these cell criteria in the early and long-term periods of cultivation after the influence of X-ray radiation at low (50 and 100 mGy) and medium doses (250 and 1000 mGy).

## 2. Results

### 2.1. MSCs Immunophenotype after X-ray Irradiation

The surface antigens phenotype of non-irradiated cells corresponded to the requirements of the International Society for Cell Therapy for MSCs: CD90^+^, CD105^+^, CD73^+^, CD45^−^, CD34^−^, HLA-DR^−^ [19], as well as CD44^+^. Further, the expression of hG-MSCs surface antigens was evaluated 9, 16, 43, and 64 days after irradiation.

There were no surface antigens expression levels changes in the studied irradiated and non-irradiated MSCs groups 9 days after irradiation (Table 1). The changes occurred in the expression levels of CD90 and CD73 surface antigens 16 days after irradiation; part of the cell population has shown a decrease in the antigens’ expression density (n = 3). For example, the CD90^dim^ population appeared in the zone of positive values in a separate cell population with a reduced fluorescence intensity (Figure 1a,b).

The correlation analysis has showed that dose-dependent changes were observed for the CD90 expression; a statistically significant expression density decrease was observed for MSCs groups which were irradiated at doses of 50 mGy, 13.3 ± 0.14% (*p* < 0.05); 100 mGy, 4.3 ± 0.21% (*p* < 0.05); 250 mGy, 4.4 ± 1.91% (*p* < 0.05); and 250 mGy, 1.2 ± 0.21% (*p* < 0.05) (Figure 1c).

The relationship between the level of CD90 expression and the radiation dose received is moderate (Cheddock scale), inverse, and linear: y = −0.00454x + 5.9113. The described pattern is reflected in the literature and may indicate that the MSCs radiosensitivity is replaced by their radioresistance with an increase in the radiation dose.

In additional to a decrease in the CD90 expression density, a less pronounced decrease in the CD105 expression density was shown for the irradiation of MSCs groups (Figure 2a,b).

As mentioned before, all the irradiated cell groups were characterized by a decrease in the CD73 expression density; the MSCs groups, which were irradiated at doses of 50 mGy, 7.0 ± 4.24% (*p* < 0.05); 100 mGy, 6.0 ± 0.78% (*p* < 0.05); and 250 mGy, 9.8 ± 0.28% (*p* < 0.05), showed the most pronounced expression decrease (Figure 3a,b). There were not any dose-dependent changes in the expression (Figure 3c).

The appearance of the CD44^dim^ cell population, which was observed 9 days after the irradiation of the MSCs group, irradiated at a dose of 50 mGy, 1.4 ± 0.23% (*p* < 0.05) (Figure 4a,b). Changes in the CD44 expression level were again observed on day 64 after irradiation; a decrease in the CD44 expression level was shown for the MSCs group which irradiated at a dose of 50 mGy, 94.5 ± 0.51% (*p* < 0.05).

### 2.2. MSCs Secretory Profile after X-ray Irradiation

Some soluble factors were quantified in the conditioned media of irradiated MSCs during long-term in vitro cultivation as part of the secretory profile study: IL-6, IL-8, IL-10, VEGF-A, and IDO. The study was conducted at several time points: 48 h, 9, 24, 43, and 64 days after irradiation. It has been shown that non-irradiated hG-MSCs do not secrete such soluble factors as IL-10 and IDO into the culture medium neither in the initial or long-term periods of cultivation. Additionally, these factors were not detected in conditioned media neither in the early or long-term periods of cultivation after the irradiation of MSCs.

On the contrary, the secretion of IL-6, IL-8, and VEGF-A was noted in the all the studied time points both in the non-irradiated control and in the irradiated groups of MSCs. An increase in the IL-6 concentration level was observed in the MSCs conditioned medium 48 h after irradiation at doses of 50 and 100 mGy—52.2 ± 2.75 pg/10^6^ cells (*p* < 0.05) and 44.0 ± 0.74 pg/10^6^ cells (*p* < 0.05), respectively—in comparison with the non-irradiated control group, 39.5 ± 0.75 pg/10^6^ cells. At the same time, a decrease in the IL-6 concentration level was observed in the MSCs conditioned medium 48 h after irradiation at doses of 250 and 1000 mGy, 35.2 ± 0.56 pg/10^6^ cells and 37.1 ± 0.00 pg/10^6^ cells, respectively (Figure 5a).

In the study of the IL-8, the significantly decrease in the IL-8 concentration in the MSCs conditioned medium was shown 48 h after irradiation at doses of 50 and 100 mGy—640.9 ± 9.60 pg/10^6^ cells (*p* < 0.05) and 544.3 ± 34.77 pg/10^6^ cells (*p* < 0.05)—respectively, in comparison with the non-irradiated control group, 694.0 ± 2.68 pg/10^6^ cells. At the same time, an increase in the IL-8 concentration was observed in the MSCs conditioned medium 48 h after irradiation at doses of 250 and 1000 mGy, 749.2 ± 12.73 pg/10^6^ cells (*p* < 0.05) and 819.1 ± 29.19 pg/10^6^ cells (*p* < 0.05) (Figure 5b).

In addition, a decrease in the VEGF-A concentration in the MSCs conditioned medium was shown through 48 h after irradiation at a dose of 50 mGy, 132.6 ± 4.09 pg/10^6^ cells (*p* < 0.05), in comparison with the non-irradiated control group, 150.9 ± 3.80 pg/10^6^ cells. At the same time, an increase in the VEGF–A concentration was observed 48 h after irradiation at doses of 100 and 250 mGy, 196.1 ± 6.91 pg/10^6^ cells (*p* < 0.05) and 203.3 ± 11.68 pg/10^6^ cells (*p* < 0.05), respectively (Figure 5c).

It Is interesting that the long-term period of cultivation demonstrated a decrease in the MSCs soluble factors concentration in the conditioned medium. For example, a significant decrease in the IL-6 concentration was observed 43 days after irradiation at doses of 250 and 1000 mGy—63.3 ± 2.85 pg/10^6^ cells (*p* < 0.05) and 52.4 ± 6.74 pg/10^6^ cells (*p* < 0.05)— respectively, in comparison with the non-irradiated control group, 76.8 ± 2.55 pg/10^6^ cells (Figure 6a). Further, a decrease in the IL-6 concentration in the conditioned medium was observed 64 days after irradiation for all the irradiated groups of MSCs, 50 mGy, 23.9 ± 3.62 pg/10^6^ cells (*p* < 0.05); 100 mGy, 28.1 ± 1.81 pg/10^6^ cells (*p* < 0.05); 250 mGy, 26.8 ± 1.26 pg/10^6^ cells (*p* < 0.05); and 1000 mGy, 27.7 ± 0.51 pg/10^6^ cells (*p* < 0.05), in comparison with the non-irradiated control group, 34.0 ± 1.04 pg/10^6^ cells (Figure 6b).

Additionally, changes were observed for the concentration of IL-8 in the conditioned media of irradiated MSCs. A decrease in the IL-8 concentration 64 days after irradiation was shown at doses of 100 mGy, 98.8 ± 4.23 pg/10^6^ cells (*p* < 0.05), 250 mGy, 95.8 ± 6.38 pg/10^6^ cells (*p* < 0.05), and 1000 mGy, 07.0 ± 2.82 pg/10^6^ cells (*p* < 0.05) in comparison with the non-irradiated control group, 132.9 ± 10.37 (Figure 6c).

### 2.3. MSCs Proliferation Activity after X-ray Irradiation

The assessment of the hG-MSCs proliferative activity was carried out 1, 9, 16, 23, 35, 43, and 64 days after irradiation; each experiment lasted for 170 h. The data normalization point of each study was 21 h after the start of the experiment.

An increase in the proliferative activity of the MSCs was noted in the study from 1 day after irradiation at a low dose of 50 mGy; a significant increase in the normalization cell index value (NCI) was noted up to 160 h after irradiation, 9.4 ± 0.36 (*p* < 0.05), compared with the non-irradiated control group, 6.4 ± 0.75 (Figure 7a). At the same time, a decrease in the proliferative activity was observed for the MSCs which were irradiated at doses of 100, 250, and 1000 mGy; a decrease in the NCI value was noted up to 120 h after irradiation at a dose of 100 mGy, 5.3 ± 0.30 (*p* < 0.05) compared with the non-irradiated control group—6.1 ± 0.34—up to 165 h after irradiation at a dose of 250 mGy, 5.0 ± 0.02 (*p* < 0.05) compared with the non-irradiated control group, 6.4 ± 0.75, and up to 145 h after irradiation at a dose of 1000 mGy, 5.2 ± 0.18 (*p* < 0.05) compared with the non-irradiated control group, 6.7 ± 0.54 (Figure 7b).

An increase in the proliferative activity of the MSCs was also noted in the study from 9 day after irradiation at a low dose of 50 mGy; an increase in the NCI value was noted up to 160 h after irradiation—7.4 ± 0.77 (*p* < 0.05)—compared with the non-irradiated control group, 4.3 ± 0.53. At the same time, the intensity of the proliferative activity of the MSCs which were irradiated at doses of 100, 250, and 1000 mGy has remained comparable to the non-irradiated control group (Figure 8).

There were no detected statistically significant differences in the proliferative activity between the non-irradiated control group and the irradiated MSCs groups in studies from 16 and 23 days after irradiation. A decrease in the proliferative activity was noted for all the irradiated groups of MSCs in the study from 35 days after irradiation. A decrease in the NCI values was observed up to 170 h after irradiation at both low and medium doses of 50 mGy, 6.3 ± 2.46 (*p* < 0.05); 100 mGy, 8.3 ± 1.46 (*p* < 0.05); 250 mGy, 5.2 ± 0.50 (*p* < 0.05); and 1000 mGy, 5.6 ± 0.24 (*p* < 0.05) in comparison with the non-irradiated control group, 17.2 ± 3.57 (Figure 8).

A decrease in the MSCs proliferative activity was noted in the study 43 days after irradiation at doses of 100, 250, and 1000 mGy; a decrease in the NCI values was noted up to 130 h after irradiation at a low dose of 100 mGy, 4.5 ± 0.58 (*p* < 0.05) compared with the control group—6.1 ± 0.83 (Figure 9a)—up to 110 h after irradiation at medium doses of 250 mGy, 3.9 ± 0.22 (*p* < 0.05) and 1000 mGy, 4.0 ± 0.20 (*p* < 0.05) in comparison to the non-irradiated control group, 4.8 ± 0.35 (Figure 9b).

There were no detected statistically significant differences in the proliferative activity between the non-irradiated control group and irradiated MSCs groups in the studies from 64 days after irradiation.

## 3. Discussion

The surface antigens CD90, CD73, and CD105 are crucial for the characterization of MSCs according to the minimum criteria put forward by the International Society for Cell Therapy [19]. It can be assumed that the observed decrease in their expression density indicates changes in the most important MSCs functional properties, the immunomodulatory properties and differentiation potencies.

CD73 (ecto-5′-nucleotidase) is a component of the adenosinergic pathway, which is key in the implementation of the MSCs immunomodulatory functions [23]. The positive expression of CD73 in tandem with ALP^+^ (alkaline phosphatase) is considered a marker of a high osteogenic differentiation potential [24]. The function of CD90 for MSCs has not been fully elucidated; however, it has been established that a CD90 expression decrease leads to an increase in MSCs differentiation potential in osteogenic and adipogenic directions in vitro [25]. Thus, there is some contradiction: on the one hand, a CD90 expression decrease indicates an increase in the MSCs differentiation potential in the adipogenic and osteogenic directions; on the other hand, a CD73 expression decrease may indicate an decrease in the osteogenic direction. However, the study of ALP^+^ was not the task of this study, and it is impossible to unambiguously discuss the uniformity of the observed patterns in the presented study and in the study of Daisy D. Canepa et al. [24]. In addition, the presented results allow only show that (based on the scattergrams of direct and lateral light scattering) a decrease in the CD90 and CD73 expression levels occurred simultaneously; there is a possibility that the detected changes indicate the appearance of two separate cell populations with an independent decrease in the CD90 and CD73 expression. There is one more contradiction in the study of Moraes DA et al. [25]: it is known that an “inverse relationship” exists between the processes of adipogenic and osteogenic differentiation in the human body; the enhancing of one process leads to the suppression of the other one [26,27]. However, the CD90^dim^ cell populations have shown also a less pronounced decrease in the CD105 expression density. This may also indicate the possibility of enhancing the MSCs differentiation potential in the adipogenic and osteogenic directions [28], and, moreover, a decrease in their functional activity as a whole [29,30]. CD44 is a hyaluronic acid receptor [31] which is involved in the processes of the migration, proliferation, adhesion, as well as the differentiation and survival of cells [32,33]. Some studies demonstrate a relationship between CD90 and CD44 expression levels; a decrease in CD90 expression leads to a decrease in CD44 expression, and this may be associated with a shift in MSCs towards a state more susceptible to differentiation [25,34]. A similar pattern was observed only for a group of cells which was irradiated at a dose of 50 mGy; however, a decrease in the CD44 and CD90 expression levels occurred in different time intervals, 9, 64, and 16 days, respectively. This may indicate that the patterns underlying the decrease in CD44 expression after irradiation differ from the ones in previous studies [25,34].

The observed changes in the factors concentration in conditioned media allow us to indirectly evaluate the changes in the MSCs functional activity that occur under the influence of radiation. Interestingly, the changes in the IL-6 (pro-inflammatory cytokine) and IL-8 (chemokine, chemoattractant) concentrations were opposite: an increase in the IL-6 concentration was accompanied by a decrease in the IL-8 concentration 48 h after irradiation at low doses of 50 and 100 mGy. At the same time, the opposite pattern was observed in the case irradiation at medium doses of 250 and 1000 mGy: a decrease in the IL-6 concentration was accompanied by an increase in the IL-8 concentration.

The revealed patterns can be explained by the involvement of NF-kB and AP-1 transcription factors in the regulation of the expression of both IL-6 and IL-8 because the promoters of these interleukin genes contain binding sites for them. Many studies have mentioned the possible involvement of these factors in the implementation of the cell radiation response [35,36,37]. The simultaneous increase in the IL-6 and IL-8 expression has been shown in the studies of radiation-induced changes in the expression and secretion under the influence of high-dose radiation [35,36,38,39]. However, there is no single traceable pattern in studies of the LDIR effects: on the one hand, researchers indicate an increase in IL-8 expression [40]; on the other hand, there are no changes in either IL–6 and IL-8 [41].

Thus, the observed patterns of IL-6 and IL-8 concentration changes under the influence of irradiation at low and medium doses can be explained: first, by the specificities of the hG-MSCs radiosensitivity, the type and power of the used radiation source, and second, by involvement in the regulation of IL-6 and IL-8 expression and secretion, an unevident factor which was activated under the influence of irradiation and led to the development of the opposite effect: stimulating and inhibiting, respectively. In general, it can be concluded that there is a difference in the inflammatory reactions implemented by the hG-MSCs under the influence of X-ray radiation at low (50 and 100 mGy) and medium (250 and 1000 mGy) doses. The results of the VEGF-A study indirectly confirm the observed patterns of the inflammatory reactions development in response to irradiation because it has an immunosuppressive effect in addition to regulating angiogenesis processes [42,43]. The concentration of VEGF-A decreased in the MSCs group was irradiated at a dose of 50 mGy (simultaneously with the increase in the IL-6 concentration and the decrease in IL-8), 48 h after irradiation. At the same time, the VEGF-A concentration which increased in the cells group was irradiated at a dose of 250 mGy (simultaneously with the decrease in the IL-6 concentration and the increase in IL-8) also 48 h after irradiation. The simultaneous increase in the IL-8 and VEGF-A concentrations can be explained by the involvement of the same regulatory factor in the regulation of their expression, for example, miR-93 [44].

In general, it is important that a decrease in IL-6 and IL-8 concentrations was shown for all irradiated groups of cells in the long-term periods of cultivation after irradiation (43 and 64 days). It can be assumed that the stimulating effect of LDIR is temporary and manifests itself only in the earliest periods of cultivation after irradiation. The decrease in IL-6 and IL-8 concentrations for all irradiated groups of MSCs in the long-term periods of cultivation may indicate a suppression of their functional immunomodulatory activity.

Additionally, X-ray radiation at a low dose of 50 mGy led to the development of the stimulating effect of the proliferative activity of hG-MSCs in the early stages of cultivation after irradiation, while doses of 100, 250, and 1000 mGy inhibit it. This phenomenon was reflected in the works of other authors who have indicated the involvement of the MAPK/ERK signaling pathway in this process [45]. The described phenomenon of the stimulating effect has faded in the long-term culture and the effects of low-radiation doses became comparable to the effects of medium radiation doses, leading to the inhibition of the MSCs proliferation. However, the effect of a low-radiation dose of 50 mGy was again noted in the study 43 days after irradiation; the irradiated cell group was more quickly aligned in the intensity of the proliferative activity with the non-irradiated control group. In the end, irradiation at both low and medium doses has no effect on the proliferative activity of the cells in the most distant period of cultivation after irradiation, as noted in the study from day 64.

The results of the presented study have indicated changes in the most important functional properties of MSCs under the influence of X-ray radiation at low (50 and 100 mGy) and medium (250 and 1000 mGy) doses, both in earlier and in the long-term periods of cultivation after exposure. In general, the observed changes in the MSCs secretory and surface antigens profiles, as well as their proliferative activity, have indicated that the effects of low and medium doses of X-ray radiation were different in the degree of severity and the direction of development in the early cultivation stages after irradiation. However, the effects of low and medium doses of X-ray radiation are comparable with each other in the long-term cultivation stages after irradiation.

## 4. Materials and Methods

### 4.1. Isolation MSCs from the Samples of Human Connective Gingiva Tissue (hG-MSCs)

The study was approved by the section of the Academic Council (extract No. 57A, dated 15 June 2021) and at the meeting of the local bioethical committee (Protocol No. 15b dated 25 June 2021) of the State Research Center, Burnasyan Federal Medical Biophysical Center of Federal Medical Biological Agency. Informed consent was obtained from all subjects involved in the study. The MSCs cultures were isolated from three male non-personalized biopsy samples of human connective gingiva tissue (8 mm^3^). The samples were incubated in a DMEM-F12 medium containing 2% fetal bovine serum, 2 mM of L-glutamine, 200 U/mL of penicillin, 200 mg/mL of streptomycin, 200 U/mL of amphorycin, and 100 U/mL of gentamicin (STEMCELL Technologies, Vancouver, BC, Canada) at 4 °C for at least 8 h. Next, the samples were dispersed and incubated in 0.25% trypsin-EDTA solution (Gibco, Waltham, MA, USA) at 37 °C for 1 h, then the reaction was stopped by adding an equal volume of FBS (Gibco, Waltham, MA, USA) and the sample was washed with 1xPBS (300 g, 7 min). Next, the samples were incubated in a 0.15% type II collagenase solution (Sigma, St.Louis, MO, USA) at 37 °C for 2 h, then the reaction was stopped by adding an equal volume of FBS and the samples were washed with 1xPBS (300 g, 7 min). The resulting cell suspensions were transferred in a 25 cm^2^ culture flask (TPP, Sweden) and cultivated in the MesenCult™ MSC Basal Medium Human culture medium with the addition of a commercial supplement (STEMCELL Technologies, Canada), antibiotics penicillin–streptomycin (50 U/mL, PanEco, Moscow, Russia) and L-glutamine (2 mM, PanEco, Moscow, Russia). The cultivation of the primary MSCs cultures was carried out for 14 days in a 5% CO_2_ incubator at 37 °C and under constant humidity.

### 4.2. Cell Line Cultivation

The cell line of hG- MSCs was characterized and standardized on the 3rd passage according to the minimum criteria of the International Society for Cell Therapy [19]. In particular, cell differentiation in three directiona (ostegennic, adipogenic, and chondogenic) was carried out (BI MSCgo^TM^ differentiation media, Israel) (Figure 10). Cultivation was carried out continuously up to and including the 12th passage. The cell line of MSCs was maintained using a culture medium MesenCult™ MSC Basal Medium Human (STEMCELL Technologies, Vancouver, BC, Canada) with the addition of a commercial supplement (STEMCELL Technologies, Vancouver, BC, Canada), antibiotics penicillin–streptomycin (50 U/mL, PanEco, Moscow, Russia) and L-glutamine (2 mM, PanEco, Moscow, Russia) under conditions of 5% of the CO_2_ incubator at 37 °C and under constant humidity. The cell line was cultured until it reached 90% confluence on the culture flask’s surface (TPP, Sweden). Then, the cells were removed from the surface of a culture flask by adding a 0.25% trypsin-EDTA solution (Gibco, Waltham, MA, USA) and incubated for 3 min at 37 °C; after that, the enzymatic reaction was stopped by adding an equal volume of FBS (Gibco, Waltham, MA, USA). Then, the surface of the flask was washed twice with a 1xPBS (Gibco, Waltham, MA, USA). The resulting cell suspension was collected in a 50 mL tube and the cells were precipitated by centrifugation (300 g, 7 min). The resulting supernatant was aspirated, and the cells was dissolved in 1xPBS. Then, the aliquot of the cells’ suspension was mixed with an equal volume of trypan blue solution and analyzed on a cell counter (Countess II Automated Cell Counter, Invitrogen) for the counting of cells and the analysis of their survival. Subsequently, the resulting cell suspension with a known concentration was sown on a new culture flask and analyzed or cryopreserved.

### 4.3. Cryopreservation and Storage of Conditioned Media

The cell-conditioned media were collected in 1.5 mL tubes (Eppendorf, Germany) and frozen at a temperature of −80 °C. Repeated freezing of the conditioned media was not allowed.

### 4.4. X-ray Irradiation

The cells were irradiated on the 4th passage in the logarithmic growth phase when they reached 70% confluence on the surface of the culture vial. Irradiation at doses of 50, 100, 250, and 1000 mGy was carried out using the X-ray unit RUST-M1 (Russia) with the specified characteristics: a dose rate of 39 mGy/min, voltage of 100 kV, aluminum filter of 1.5 mm, and at a temperature of 4 °C. The error of the emitted dose did not exceed 15%. The cells were cultured continuously for 64 days after irradiation. The non-irradiated control group of MSCs was cultured and analyzed in parallel with the irradiated cell groups throughout the study. According to the recommendations of the ICRP, the doses of 50 and 100 mGy belong to the range of low-radiation doses and doses of 250 and 1000 mGy belong to the range of medium-radiation doses [1].

### 4.5. Immunophenotyping of Cells

The level of MSCs surface antigens was assessed using a flow cytometer BD FACS Canto II (Becton Dickinson, Franklin Lakes, NJ, USA) with two lasers: 488 nm and 633 nm. The calibration of the device before the measurements was carried out using the commercial Cytometer Setup and Tracking Beads Kit (BD, Franklin Lakes, NJ, USA), as well as BD CompBeads Anti-Mouse Ig (BD, Franklin Lakes, NJ, USA) for compensating the protocol settings. The cells were stained with fluorescently labeled antibodies and dyes (BD, Franklin Lakes, NJ, USA) according to the presented panel (Table 2). The obtained results were processed using FlowJo (BD, Franklin Lakes, NJ, USA) software.

The expression of MSCs surface antigens was evaluated 9, 16, 43, and 64 days after irradiation. There was one experiment conducted for each donor at each time point (n = 3, three MSCs donors).

### 4.6. The Enzyme-Linked Immunosorbent Assay (ELISA)

An ELISA of conditioned media soluble factors was carried out using commercial kits:Human IL-8/CXCL8 Quantikine Elisa Kit (#D8000C, R&D systems a bio-techne brand, Minneapolis, MN, USA);Human IL-6 Elisa Kit (#BMS213-2, Invitrogen, Waltham, MA, USA);Human IL10 Elisa kit (#BMS215-2, Invitrogen, Waltham, MA, USA);Human VEGF-A Elisa kit (#BMS277-2, Invitrogen, Waltham, MA, USA);Human IDO Elisa kit (#EH246RB, Invitrogen, Waltham, MA, USA).

The results were recorded by the iMark (Bio-Rad, USA) reader using the Zemfira software. The results were presented taking into consideration the number of cells and volume of the condition medium in the culture flask at the time of the conditioned medium’s collection (pg/10^6^ cells). The study was conducted at several time points: 48 h, 9, 24, 43, and 64 days after irradiation. There were two repetitions for each donor at each time point (n = 6). The intra-assay coefficients of variability (n = 25) were less than 10: 9.8 for IL-6, 5.6 for IL-8, and 9.3 for VEGF-A. The inter-assay coefficients of variability (n = 2) were less than 15: 8.4 for IL-6, 7.7 for IL-8, and 13.8 for VEGF-A.

### 4.7. Evaluation of Cell Proliferative Activity

The proliferative activity of the cells was evaluated using the xCelligence RTCA cell analyzer (ACEA Biosciences, Santa Clara, CA, USA), which was installed in a CO_2_ incubator. It allowed a real-time evaluation of the cell’s proliferative activity due to the detection of the cellular index (CI) (electrical resistance at the bottom of well covered with gold). The suspension of the cells was sown into the wells of commercial E-plate 16 culture plates (ACEA Biosciences, Santa Clara, CA, USA) in the amount of 1500 cells/well; after that, the CI values were monitored in real time. The results were presented as a normalized cell index (NCI), which was calculated using xCelligence RTCA software: the CI values at a certain time point—the point of normalization (for example, when cells were stimulated)—was set by the software as 1.0. All subsequent values of CI were represented as a proportion to CI at the point of normalization. The assessment of the MSCs proliferative activity was carried out with 1, 9, 16, 23, 35, 43, and 64 days after irradiation; each study lasted for 170 h. The data normalization point of each study was 21 h after the start of the experiment. There was one experiment for each donor at each time point (n = 3).

### 4.8. Statistics

Each of the irradiated cell groups was compared with the non-irradiated control group of MSCs at each studied time point. Statistical processing of the results was carried out using the Statistica 6.0 software (Statsoft, Tusla, OK, USA). The significance of the differences was assessed using the paired sample *t*-test. The results were presented as the arithmetic mean and at least three independent experiments ± standard deviation. The differences were considered statistically significant at *p* < 0.05.

## 5. Conclusions

The observed changes in the MSCs immunological profile at the earliest stages of cultivation after irradiation (9 days) have clearly demonstrated the development of a stimulating effect under the influence of X-ray radiation at 50, 100, and 250 mGy. We suppose that this phenomenon reflects the reversing the cell radiosensitivity by their radioresistance, and has been expressed in a decrease in the CD90, CD73, and CD105 expression level. Additionally, it can be assumed that the observed changes in the surface antigens expression level allows us to indirectly discuss the changes in the MSCs functional activity, in particular, their differentiation potential in the adipogenic and osteogenic directions, occurring under the influence of X-ray radiation at low and medium doses. Thus, the MSCs surface antigens can act as a promising criterion for assessing the risks of exposure to radiation at low and medium doses on the human body due to the availability, standardization, and accuracy of the flow cytometry, as well as the observed dose-dependent changes in the CD90 expression level under the influence of X-ray radiation. Of course, it remains to determine the exact functional causes of the observed changes, to supplement the studied panel of MSCs surface antigens, as well as to evaluate the observed expression changes in the context of the accumulation and repair of double-stranded DNA breaks.

The study of the MSCs secretory profile also has demonstrated the development of stimulating effects at the earliest stages of cultivation after irradiation at low doses. The irradiation at low doses of 50 and 100 mGy and medium doses of 250 and 1000 mGy has led to the development of opposite inflammatory reactions: an increase in the IL-6 concentration (pro-inflammatory cytokine) and a decrease in the IL-8 concentration (chemokine) in a conditioned medium after irradiation at doses of 50 and 100 mGy, while the opposite pattern was observed after irradiation at medium doses: a decrease in the IL-6 concentration and an increase in the IL-8 concentration. It is important that a decrease in IL-6 and IL-8 concentrations was observed for all irradiated MSCs groups in the long-term periods of cultivation after irradiation. We argue that these changes have indicated a decrease in the functional and immunomodulatory activity of MSCs.

The stimulating effect of irradiation at low doses was again demonstrated in the study of the MSCs proliferative activity: a group of cells irradiated at a dose of 50 mGy has showed an increase in proliferative activity in the early stages of cultivation after irradiation (from 1 to 15 days). However, the proliferative activity of all the irradiated MSCs groups has decreased in comparison with the non-irradiated control group in the long-term culture. We argue that this indicates a decrease in the MSCs functional activity in addition to the previously described decrease in the IL-6 and IL-8 concentrations.

In general, the study of the changes which occur in the MSCs functional activity under the influence of X-ray radiation at low and medium doses is especially important because MSCs play the role of a regenerative reserve in the human body and have the ability to self-sustain and the potential to differentiate. The study of the secretory and immunology profile, as well as the proliferative activity of MSCs, allows us to discuss the changes in their functional properties. Some of these can be considered promising criteria for assessing the risks of an exposure to low and medium doses because dose-dependent changes were shown. At the same time, the hG-MSCs, due to their biological accessibility (low invasiveness of the biomaterial obtaining procedure), simplicity of isolation, and stability of the cell line, seem to be a convenient model for studying the effects of irradiation, and in the future, it may probably be proposed to assess individual human radiosensitivity, in particular, of people who work in the nuclear industry.

## Figures and Tables

**Figure 1 ijms-24-06346-f001:**
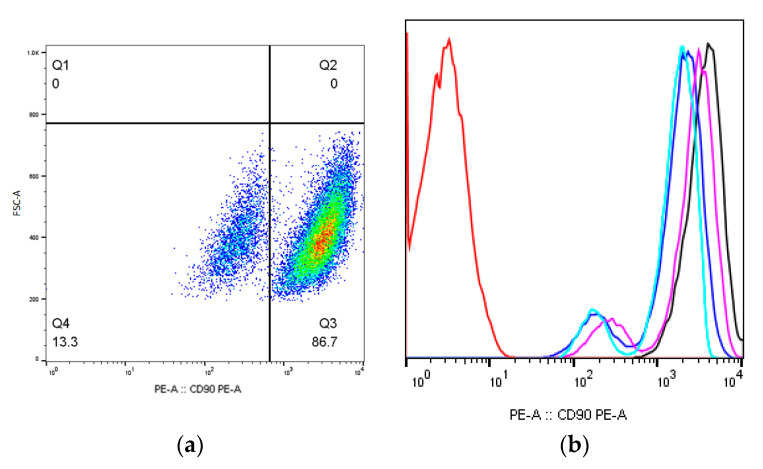
The CD90 expression level one of the donors MSCs 16 days after irradiation at dose of 50 mGy: (**a**) pseudocolor scattergram of stained CD90^dim^ and CD90^+^ population; (**b**) histogram of unstained control (red), stained non-irradiated control (black) and CD90^dim^ and CD90^+^ populations of stained irradiated MSCs from 3 donors, n = 3 (pink, blue, and light blue); (**c**) the count of CD90^dim^ cells in non-irradiated and irradiated populations of MSCs (n = 3) 16 days after irradiation at low (50 and 100 mGy) and medium (250 and 1000 mGy) doses. * (*p* < 0.05).

**Figure 2 ijms-24-06346-f002:**
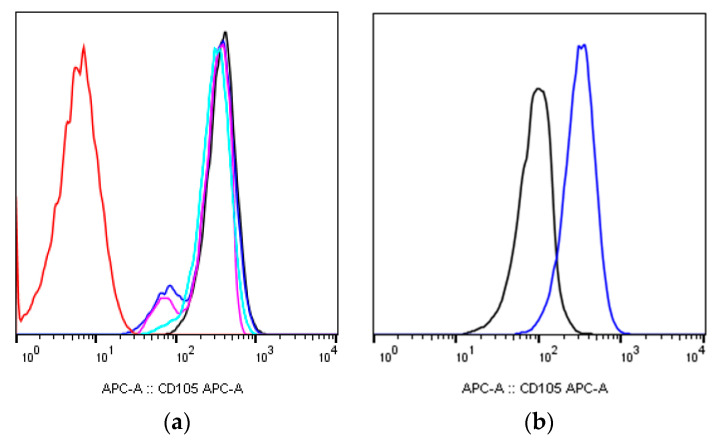
The CD105 expression level one of the donors MSCs 16 days after irradiation at dose of 50 mGy: (**a**) histogram of unstained control (red), stained non-irradiated control (black), and CD105^dim^ and CD105^+^ populations of stained irradiated MSCs from 3 donors, n = 3 (pink, blue, and light blue); (**b**) histogram of difference in fluorescence between CD90^+^/CD105^dim^ (black) and CD90^+^/CD105^+^ populations (blue).

**Figure 3 ijms-24-06346-f003:**
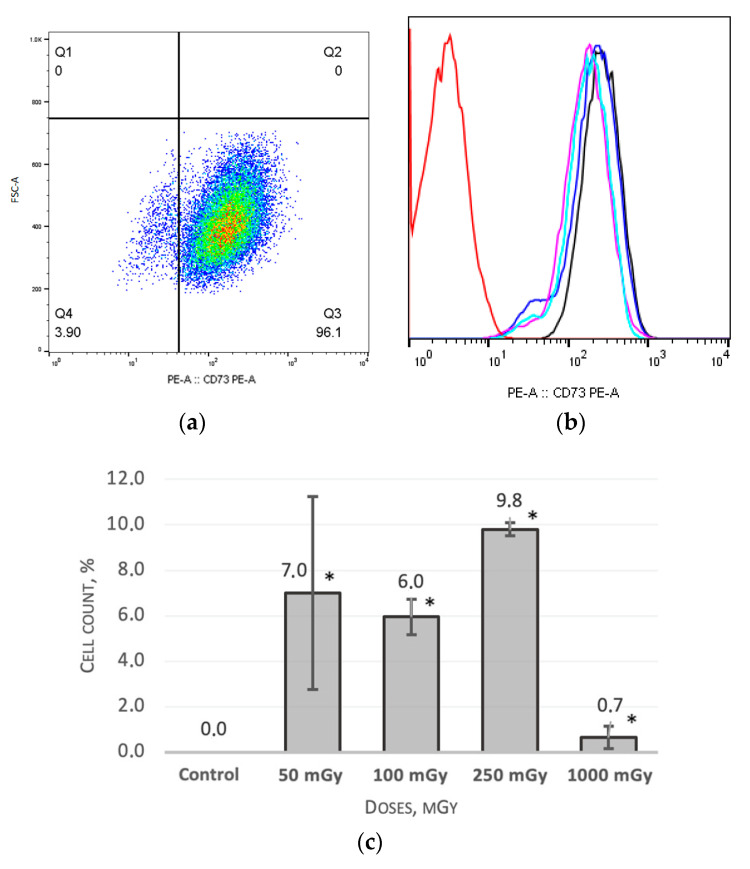
The CD73 expression level one of the donors MSCs 16 days after irradiation at dose of 50 mGy: (**a**) pseudocolor scattergram of stained CD73^dim^ and CD73^+^ population; (**b**) histogram of unstained control (red), stained non-irradiated control (black), and CD73^dim^ and CD73^+^ populations of stained irradiated MSCs from 3 donors, n = 3 (pink, blue, and light blue); (**c**) the count of CD73^dim^ cells in non-irradiated and irradiated populations of MSCs (n = 3) 16 days after irradiation at low (50 and 100 mGy) and medium (250 and 1000 mGy) doses. * (*p* < 0.05).

**Figure 4 ijms-24-06346-f004:**
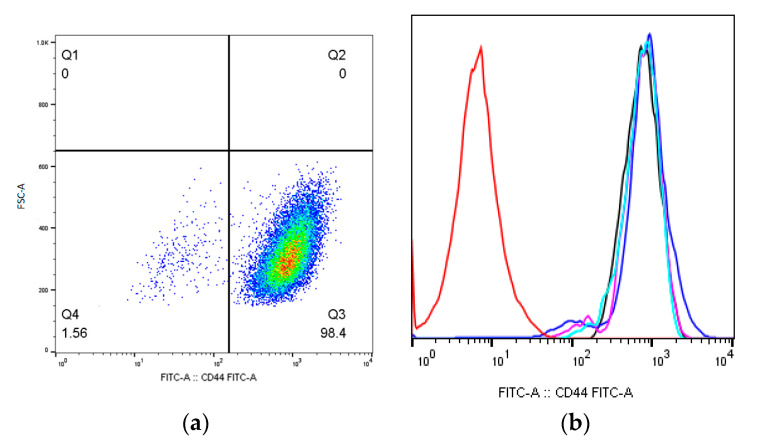
The CD44 expression level one of the donors MSCs 9 days after irradiation at dose of 50 mGy: (**a**) pseudocolor scattergram of stained CD44^dim^ and CD44^+^ population; (**b**) histogram of unstained control (red), stained non-irradiated control (black), and CD44^dim^ and CD44^+^ population of stained irradiated MSCs from 3 donors, n = 3 (pink, blue, and light blue).

**Figure 5 ijms-24-06346-f005:**
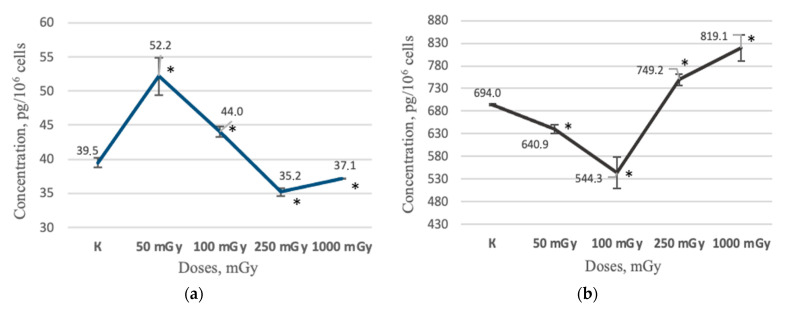
The results of ELISA test of conditioned mediums non-irradiated and irradiated MSCs groups (n = 6): (**a**) the IL-6 concentration in MSCs conditioned medium 48 h after irradiation; (**b**) the IL-8 concentration in MSCs conditioned medium 48 h after irradiation; (**c**) the VEGF-A concentration in MSCs conditioned medium 48 h after irradiation. * (*p* < 0.05).

**Figure 6 ijms-24-06346-f006:**
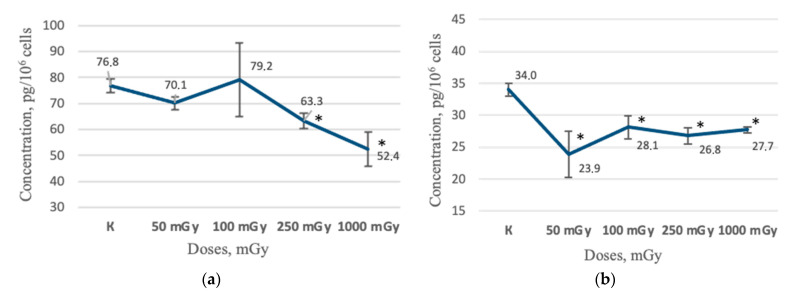
The results of ELISA test of conditioned mediums non-irradiated and irradiated MSCs groups (n = 6): (**a**) the IL-6 concentration in MSCs conditioned medium 43 days after irradiation; (**b**) the IL-6 concentration in MSCs conditioned medium 64 days after irradiation; (**c**) the IL-8 concentration in MSCs conditioned medium through 64 days after irradiation. * (*p* < 0.05).

**Figure 7 ijms-24-06346-f007:**
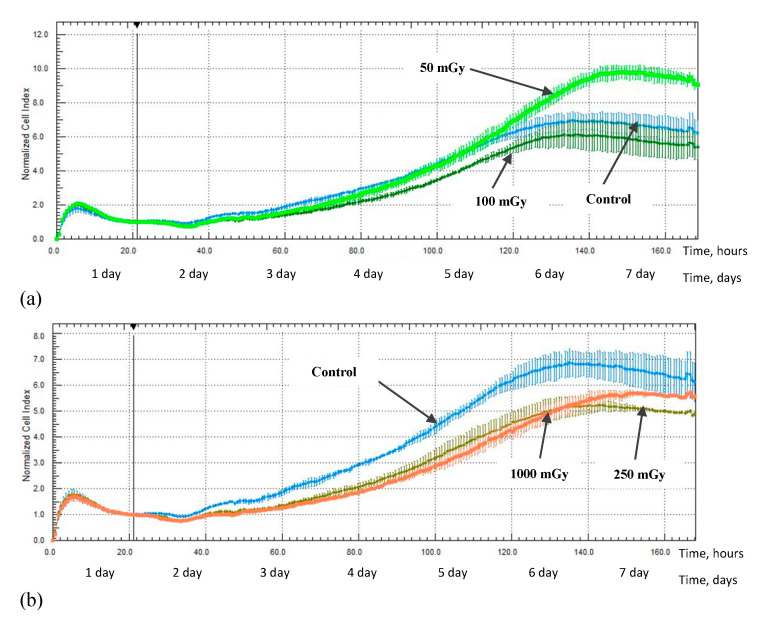
The proliferative activity of MSCs (n = 3) from 1 day after irradiation at doses of: (**a**) 50 mGy (light green) and 100 mGy (dark green), (**b**) 250 mGy (green) and 1000 mGy (orange).

**Figure 8 ijms-24-06346-f008:**
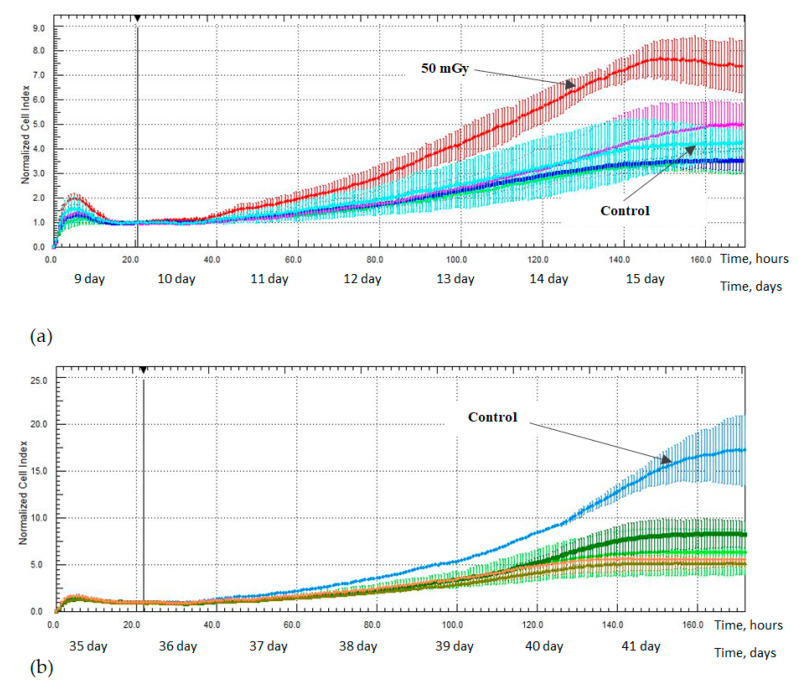
The proliferative activity of MSCs (n = 3): (**a**) from 9 days after irradiation at doses of: 50 mGy (red) and 100 mGy (pink), 250 mGy (green) and 1000 mGy (blue); (**b**) from 35 days after irradiation at doses of: 50 mGy (light green) and 100 mGy (dark green), 250 mGy (green) and 1000 mGy (orange).

**Figure 9 ijms-24-06346-f009:**
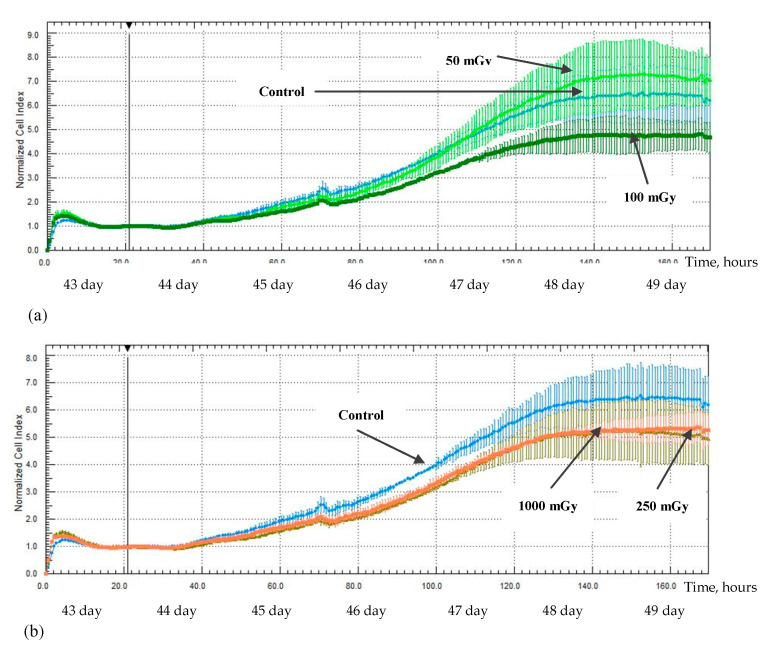
The proliferative activity of MSCs (n = 3) from 43 days after irradiation at doses of: (**a**) 50 mGy (light green) and 100 mGy (dark green), (**b**) 250 mGy (green) and 1000 mGy (orange).

**Figure 10 ijms-24-06346-f010:**
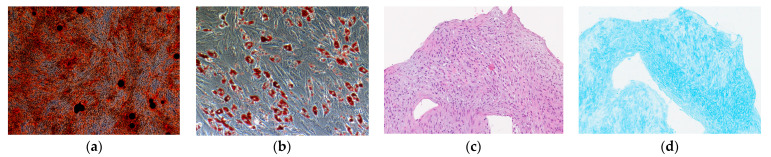
The results of hG-MSCs differentiation in three directions: (**a**) osteogenic (100×, ARS), (**b**) adipogenic (200×, Oil Red O), (**c**) chondrogenic (200×, results of histology: immature mesenchymal tissue with a tendency to form cartilage tissue), (**d**) chondrogenic (200×, Alcian Blue).

**Table 1 ijms-24-06346-t001:** The results of MSCs immunophenotyping 9 days after irradiation.

Parameters	K	50 mGy	100 mGy	250 mGy	1000 mGy
	% of Positive Cells				
CD45	0.0 ± 0.00	0.0 ± 0.00	0.0 ± 0.00	0.0 ± 0.00	0.0 ± 0.00
CD90	99.9 ± 0.00	99.9 ± 0.20	99.9 ± 0.12	99.9 ± 0.00	99.9 ± 0.08
CD105	99.9 ± 0.15	99.9 ± 0.24	99.9 ± 0.21	99.9 ± 0.10	99.9 ± 0.00
CD34	0.0 ± 0.00	0.0 ± 0.00	0.0 ± 0.00	0.0 ± 0.00	0.0 ± 0.00
CD73	99.9 ± 0.00	99.9 ± 0.00	99.9 ± 0.23	99.9 ± 0.33	99.9 ± 0.19
HLA-DR	0.0 ± 0.00	0.0 ± 0.00	0.0 ± 0.00	0.0 ± 0.00	0.0 ± 0.00
CD44	99.9 ± 0.00	98.6 ± 0.20	99.9 ± 0.36	99.9 ± 0.16	99.9 ± 0.31

**Table 2 ijms-24-06346-t002:** The panel of used fluorescently labeled antibodies and dyes.

Tube 1	Tube 2	Tube 3	Tube 4	Tube 5
Unstained control	7AAD	CD45 FITC	CD34 FITC	CD44 FITC
		CD90 PE	CD73 PE	-
		CD105 APC	HLA-DR APC	-

## Data Availability

The data presented in this study are available on request from the corresponding author.

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
