# Peer review of "Evaluation of Changes in Some Functional Properties of Human Mesenchymal Stromal Cells Induced by Low Doses of Ionizing Radiation"

_ijms, 2023, doi:10.3390/ijms24076346_

Round 1
Reviewer 1 Report
The study aimed to evaluate the effect of low and medium doses irradiation on chosen properties of human MSCs isolated from gum connective tissue. The idea is simple, but the study could be potentially interesting and valuable. Unfortunately, it contains a numer a serious flaws and should not be published it this or similar form in my opinion. The major problems are:
1. Description of methods does not clearly state how many donors were used in this study. There is a suggestion that only one (singular form of a word „sample”). If so, the results of the study does not represent proper scientific quality and this is the most important problem of this manuscript. Drawing the conclusions based on the analysis of only one primary population (especially when unusual MSC source is used) is not justified.
2. There is no data regarding the differentiation potential of isolated population.
3. the method desription is missing many details (see the comments in the PDF file)
4. the presentation of data is pretty messy. No information about „n”, statistical methods, controls, graph axes cut etc. (Detailed comments in PDF file)
There are numerous minor points included in comments in PDF file
The manuscript also would require careful linguistic proofreading. In the PDF file, I have indicated several sentences that need to be corrected, but there are definitely more than those pointed out.
The whole backmatter, including the description of ethical issues, is missing

Author Response
Сover letter #1
Dear Reviewer,
Thank you for comments and notices on our mini-review. And we clarify following questions:
- Question: Description of methods does not clearly state how many donors were used in this study. There is a suggestion that only one (singular form of a word „sample”). If so, the results of the study do not represent proper scientific quality and this is the most important problem of this manuscript. Drawing the conclusions based on the analysis of only one primary population (especially when unusual MSC source is used) is not justified.
Answer: We have used MSCs of three male donors in this study. The necessary changes were made in the text of article.
- Question: There is no data regarding the differentiation potential of isolated population.
Answer: It was noted at point 4.2. – Cell line cultivation: “The cell line of MSCs human connective gingiva tissue was characterized and standardized on the 3rd passage according to the minimum criteria of the International Society for Cell Therapy”. We carried out differentiation of donor MSCs to three directions and confirmed their properties, but the study of cell differentiation was not the purpose of our study, and more specific information about the results of differentiation of MSCs was not provided.
- Question: The method description is missing many details (see the comments in the PDF file).
Answer:
4.1. Isolation MSCs from the sample of human connective gum tissue
- The statement referring to the ethical issues was added.
- There were three samples. The necessary changes were made in the text.
4.2. Cell line cultivation
- It was noted at point 2. – Cell line cultivation: “The cell line of MSCs human connective gingiva tissue was characterized and standardized on the 3rd passage according to the minimum criteria of the International Society for Cell Therapy”. We carried out differentiation of donor MSCs to three directions and confirmed their properties, but the study of cell differentiation was not the purpose of our study, and more specific information about the results of differentiation of MSCs was not provided.
4.4. X-ray irradiation
- The dose rate was 39 mGy/min. Thus the irradiation time was differed for each dose – 50, 100, 250 and 1000 mGy. The non-irradiated control group of MSCs was cultured and analyzed in parallel with the irradiated cell groups throughout the study. We have noted it in the methods. The time point of the cell collected/analysed were presented in the text of article. Additional we have noted it in the methods.
4.6. The enzyme-linked immunosorbent assay (ELISA)
- The aliquots of conditioned media from each group of cells were collected in Eppendorf at the specified time point (5 repetitions, 1 for each studied factor). Data of the collected medium total volume from the flasks and the number of cells at the time of collection were not presented, however the results were presented taking into account the number of cells and the volume of the medium in the culture flask at the time of collecting the medium (pg/106 cells). This was noted in the text of the article. The collection of media for all the studied groups - the non-irradiated control group and the irradiated one - was carried out simultaneously. There were two repetitions for each donor at each point in time. The method of cell counting after medium collection was noted at point 2. - Cell line cultivation: “Then aliquot of cells suspension was mixed with equal volume of trypan blue solution and analyzed on a cell counter (Countess II Automated Cell Counter, Invitrogen) for counting of cells and analyze of their survival. Subsequently, the resulting cell suspension with a known concentration was sown on a new culture flask, analyzed or cryopreserved.”
4.8. Statistics
The distribution of data in each sample correspond to the normal distribution. Each of the irradiated cell groups as compared with the nonirradiated control group at each studied time point.
- The information about “n” was added for all experiments.
4. Question: The presentation of data is pretty messy. No information about „n”, statistical methods, controls, graph axes cut etc. (Detailed comments in PDF file)
Answer:
- The more concretely information about non-irradiated control group and “n” was added in the text of article.
- We have used the Paired Sample T-test.
- Question: There are numerous minor points included in comments in PDF file
Answer:
- The title of article was corrected.
- The fluorescence intensity of the unpainted control group of cells is indicated on histograms №1b, 3b and 4b.
- The cells were cultured continuously for 64 days after irradiation. The non-irradiated control group of MSCs was cultured and analysed in parallel with the irradiated cell groups throughout the study.
- The irradiated cell groups were compared with the non-irradiated control group. The data were illustrated on graphs and the values of the signs of the control group were also addeded in the text of article.
- The abbreviation hG-MSCs was adopted in the text of the article.
- The figure captions have been corrected.
- The results of IDA and IL-10 ELISA test of conditioned medium nonirradiated and irradiated MSCs groups were obtained by us independently.
- The Y axis was started from 0 point for reader convenience.
- Question: The manuscript also would require careful linguistic proofreading. In the PDF file, I have indicated several sentences that need to be corrected, but there are definitely more than those pointed out.
Answer: The article will be corrected before publication in the editorial office of the journal.
- Question: The whole backmatter, including the description of ethical issues, is missing
Answer: The materials were added in the text of article.
With best regards,
Authors.

Reviewer 2 Report
International Journal of Molecular Sciences
COMMENTS TO THE EDITORS AND THE AUTHORS
Dear the Editor and the Authors,
Please find enclosed the comments for the above-mentioned manuscript.
A SUMMARY OF THE CONTENT
The authors stated that the study investigated the effects of irradiation by low doses of X-ray radiation on the model of MSCs of human connective gum tissue. The results showed changes in the expressions of MSCs surface antigens CD90, CD73, CD105 and CD44 changes under the influence of X-ray radiation (50, 100 and 250 mGy) in the early stages of cultivation after irradiation. Besides, the most important functional characteristics of MSCs such as the stemness degree and differentiation potencies where changed. The low (50 and 100 mGy) and medium (250 and 1000 mGy) doses of X-ray radiation lead to the development of opposite inflammatory reactions in the early stages of cultivation after irradiation. The decreased concentrations of the interleukins concentrations were observed in all irradiation groups of cells in the long-term periods of cultivation after irradiation. The low dose (50 mGy) increased the MSCs proliferative activity in the early stages of cultivation after irradiation, while the proliferative activity of all irradiated cell groups decreased in the long-term periods of cultivation after irradiation. The authors concluded that the results of the immunological and secretory profiles, as well as the cell proliferative activity of MSCs have indicated development of stimulating effects in the early stages of cultivation after irradiation by low doses of X-ray radiation. On the contrary the effects of low doses were comparable with the effects of medium doses of X-ray radiation in the long-term periods of cultivation after irradiation and have indicated inhibition of the MSCs functional activity of cells.
THE OVERALL OPINION OF THE MANUSCRIPT
The strengths: the manuscript presents new knowledge; the results were obtained using the human cells; the figures very clearly present the results.
The limitations: the title is not precisely formulated; the sex-related differences in the responses are not presented; the mechanistic approach is missing; the citation in the introduction and the discussion of the original and the important pioneered results, as well as recent advance in the field focusing on the subject of the study is missing; the intra- and inter- assay coefficients are not presented; the Fundings Statement is missing; the Institutional Review Board Statement is missing; the Informed Consent Statement is missing; the Data Availability Statement is missing, the Conflicts of Interest Statement is missing.
Accordingly, major revision is required for the further consideration.
Please find enclosed some of the suggestions in the comments to the authors listed below.
(1) TITLE
Please consider modifying the title to precisely reflect the results presented in the manuscript. Namely, the results do not show the whole “secretory and immunology profiles”.
(2) ABSTRACT
2.1. Please avoid describing the background in many sentences (1/3 of the text of the abstract describe the background) and please focus on the aim, the methodology and the results of the study.
2.2. Please provide results related to the differences between responses of human MSC obtained from females and males.
2.3. Please provide results related to the mechanistic approach.
2.4. Please avoid citation of the references.
(3) INTRODUCTION
3.1. Please describe the original, and the important pioneered results, as well as recent advance in the field focusing on the subject of the study.
(4) MATERIALS AND METHODS
4.1. Please provide the mechanistic studies to prove the physiological significance.
4.2. Please provide the studies related to sex-differences i.e. the studies related to the differences in the responses of human MSC obtained from females and males.
4.3. Please provide intra- as well as inter-assay coefficients for all analyses.
(5) RESULTS
5.1. Please provide the results obtained from the new experiments related to the mechanistic approach.
5.2. Please provide the results related to the differences in the responses of human MSC obtained from females and males.
(6) DISCUSSION
6.1. Please discuss the original, and important pioneered results, as well as recent advance in the field focusing on the subject of the study.
6.2. Please discuss the new results related to the mechanistic approach.
6.3. Please discuss the results related to the differences in the responses of human MSC obtained from females and males.
(7) REFERENCES
7.1. Please provide references describing the original, and important and pioneered results, but also references describing the recent advance in the field.
(8) FIGURES and FIGURE LEGENDS
8.1. Please provide the new figures and figure legends showing the new results related to the mechanistic approach.
8.2. Please provide the new figures and figure legends showing the new the results related to the differences in the responses of human MSC obtained from females and males.
(9) GENERAL
Please provide following the statements listed below.
9.1.The Fundings Statement.
9.2. The Institutional Review Board Statement.
9.3. The Informed Consent Statement.
9.4. The Data Availability Statement is missing.
9.5. The Conflicts of Interest Statement is missing.
9.6. Please use the format of the chapters and the paragraphs required for the IJMS.
Good luck and all the best J
Author Response
Сover letter #2
Dear Reviewer,
Thank you for comments and notices on our mini-review. And we clarify following questions:
- TITLE
Question: Please consider modifying the title to precisely reflect the results presented in the manuscript. Namely, the results do not show the whole “secretory and immunology profiles”.
Answer: The title of article was corrected.
- ABSTRACT
Question:
- Please avoid describing the background in many sentences (1/3 of the text of the abstract describe the background) and please focus on the aim, the methodology and the results of the study.
- Please provide results related to the differences between responses of human MSC obtained from females and males.
- Please provide results related to the mechanistic approach.
- Please avoid citation of the references.
Answer: The abstract was corrected. The MSCs cultures were isolated from three male non-personalized biopsy samples of human connective gingiva tissue.
The presented results of MSCs secretory profile and immunophenotype changes under the influence of low radiation doses do not seem sufficient for conducting a more extensive mechanical approach. Today the IRCP still doubts the validity of the threshold or non-threshold concept and we cannot draw serious conclusions based on the data obtained in vitro. However, the observed dose-dependent changes of CD90 expression were analyzed in a similar way.
- INTRODUCTION
Question:
- Please describe the original, and the important pioneered results, as well as recent advance in the field focusing on the subject of the study.
Answer: We have studied MSCs from human connective gingival tissue. It is not popular source of MSCs and we have based on the results of studies which have used other source of cells and irradiation.
- MATERIALS AND METHODS
Question:
4.1. Please provide the mechanistic studies to prove the physiological significance.
4.2. Please provide the studies related to sex-differences i.e. the studies related to the differences in the responses of human MSC obtained from females and males.
4.3. Please provide intra- as well as inter-assay coefficients for all analyses.
Answer: Please give more concretely explanation intra- as well as inter-assay coefficients.
- RESULTS
Question:
5.1. Please provide the results obtained from the new experiments related to the mechanistic approach.
5.2. Please provide the results related to the differences in the responses of human MSC obtained from females and males.
Answer: Please see the answers above.
- DISCUSSION
Question:
6.1. Please discuss the original, and important pioneered results, as well as recent advance in the field focusing on the subject of the study.
6.2. Please discuss the new results related to the mechanistic approach.
6.3. Please discuss the results related to the differences in the responses of human MSC obtained from females and males.
Answer: Please see the answers above.
- REFERENCES
Question:
- Please provide references describing the original, and important and pioneered results, but also references describing the recent advance in the field.
Answer: Please see the answers above.
- FIGURES and FIGURE LEGENDS
Question:
8.1. Please provide the new figures and figure legends showing the new results related to the mechanistic approach.
8.2. Please provide the new figures and figure legends showing the new the results related to the differences in the responses of human MSC obtained from females and males.
Answer: Please see the answers above.
- GENERAL
Question:
Please provide following the statements listed below.
9.1. The Fundings Statement.
9.2. Institutional Review Board Statement.
9.3. The Informed Consent Statement.
9.4. The Data Availability Statement is missing.
9.5. The Conflicts of Interest Statement is missing.
9.6. Please use the format of the chapters and the paragraphs required for the IJMS.
Answer: This information was added in the text of article.
With best regards,
Authors.

Reviewer 3 Report
In this study, authors investigated the effect of radiation on MSC cell surface markers, secreted factors and proliferation capacity. Below are my comments/suggestions:
- Line 112: Authors noted that the expression of cell surface markers were not changed at day 9 time point, but they didn’t include any data. Flow cytometry data (MFI and %of positive cells for each markers should be included).
- Line 115: Authors noted in line 115 that n=3 for that experiment.Could authors clarify if they performed the experiment on 3 MSC donors or 3 technical replicates from the same donor?
- Fig 1, a,b: Although it’s acceptable to pick a histogram from one of the replicates as a representative histogram, authors should have included a graph from the three replicates for MFI and frequency of positive cells for the given markers, followed by statistical analysis. It would be helpful to add the number of replicates in the figure legend too.
- Fig 3:Y axis labels missing. It’s not clear what each of black and blue represents. This graph is not explained clearly in the text too.
Fig 2,3 and 6: Similar to my note for fig 1, MFI data from different donors should be graphed to show the donor response heterogeneity.
- Data from fig 1 to Fig 6 are considered as a group of related experiments, there is no need to show each panel as a separate figures. These 6 figures may be merged as 1-2 figures, this will help to follow the results better.
- Fig 7 and 8 may be combined
- Fig 9 and 10 may be combined
- Fig 14: This figure is not explained in the result section at all!
- Did author tested if radiation affects the three lineage differentiation of MSCs
- Conclusion section is too long and overlaps with the discussion section. Shortening the conclusion section is recommended.
- There are grammar error throughout the manuscripts.
Author Response
Dear Reviewer,
Thank you for your attention for details. I have prepared answers for questions:
- Line 112: Authors noted that the expression of cell surface markers were not changed at day 9 time point, but they didn’t include any data. Flow cytometry data (MFI and % of positive cells for each markers should be included).
Answer:
Table 1. The results of MSCs immunofenotyping through 9 days after irradiation.
|
Parameters |
К |
50 mGy |
100 mGy |
250 mGy |
1000 mGy |
|
% of positive cells |
|||||
|
CD45 |
0,0±0,00 |
0,0±0,00 |
0,0±0,00 |
0,0±0,00 |
0,0±0,00 |
|
CD90 |
99,9±0,00 |
99,9±0,20 |
99,9±0,12 |
99,9±0,00 |
99,9±0,08 |
|
CD105 |
99,9±0,15 |
99,9±0,24 |
99,9±0,21 |
99,9±0,10 |
99,9±0,00 |
|
CD34 |
0,0±0,00 |
0,0±0,00 |
0,0±0,00 |
0,0±0,00 |
0,0±0,00 |
|
CD73 |
99,9±0,00 |
99,9±0,00 |
99,9±0,23 |
99,9±0,33 |
99,9±0,19 |
|
HLA-DR |
0,0±0,00 |
0,0±0,00 |
0,0±0,00 |
0,0±0,00 |
0,0±0,00 |
|
CD44 |
99,9±0,00 |
98,6±0,20 |
99,9±0,36 |
99,9±0,16 |
99,9±0,31 |
- Line 115: Authors noted in line 115 that n=3 for that experiment. Could authors clarify if they performed the experiment on 3 MSC donors or 3 technical replicates from the same donor?
Answer: The experiment was performed on 3 MSCs donors. The information was added in materials and methods (4.5. Immunophenotyping of cells).
- Fig 1, a,b: Although it’s acceptable to pick a histogram from one of the replicates as a representative histogram, authors should have included a graph from the three replicates for MFI and frequency of positive cells for the given markers, followed by statistical analysis. It would be helpful to add the number of replicates in the figure legend too.
Answer:
The information about MFI and frequency of positive cells was added on the figures 1,3,4,6.
- Fig 3: Y axis labels missing. It’s not clear what each of black and blue represents. This graph is not explained clearly in the text too.
Answer: Y axis label was added. The color designations are indicated in the description of the figure.
- Data from fig 1 to Fig 6 are considered as a group of related experiments, there is no need to show each panel as a separate figures. These 6 figures may be merged as 1-2 figures, this will help to follow the results better.
Answer: It was corrected.
- Fig 7 and 8 may be combined
Answer: It was corrected.
- Fig 9 and 10 may be combined
Answer: It was corrected.
- Fig 14: This figure is not explained in the result section at all!
Answer:
4.2. Cell line cultivation
The cell line of hG-MSCs was characterized and standardized on the 3rd passage according to the minimum criteria of the International Society for Cell Therapy [19]. In particular, cell differentiation in three direction (ostegennic, adipogenic and hondogenic) was carry out (BI MSCgoTM differentiation media, Israel) (Figure 14).
- Did author tested if radiation affects the three lineage differentiation of MSCs
Answer: No, it was not the task of researching.
- Conclusion section is too long and overlaps with the discussion section. Shortening the conclusion section is recommended.
Answer: Conclusion section was previously edited in accordance with the comments of another reviewer. If possible, we would like to keep this volume of material.
- There are grammar error throughout the manuscripts.
Answer: The article will be corrected before publication in the editorial office of the journal.
Best regards, Authors

Round 2
Reviewer 1 Report
Dear Authors,
The manuscript has been substantially improved, but still several issues would require correction. These are:
If you mention ISCT requirments you should show 3-lineage differentiation potential (or cite your previous paper in which it was demonstrated). If these data cannot be shown (because of unknown reasons) - it is not justified to say the cells were characterized according to the ISCT requirments. Gingiva is relatively uncommon source of MSCs and therefore differentiation potential should be shown.
Flow cytometry plots/histograms from control (stained, non-irradiated cells) should be added to the figures.
Mean normalized cell index from all 3 donors should be presented in a graphical way.
The mFISH technique (mentioned once in the method section) is neither described in method section (as separate method) nor any data are presented in results. So it should be either expanded or deleted.
The y axis still do not start from 0 in graphs showing secretion.
As previosly, there are numerous minor points indicated in PDF file comments (attached).
Author Response
Dear Reviewer,
Thank you for your attention for details. I have prepared answers for questions:
- If you mention ISCT requirments you should show 3-lineage differentiation potential (or cite your previous paper in which it was demonstrated). If these data cannot be shown (because of unknown reasons) - it is not justified to say the cells were characterized according to the ISCT requirments. Gingiva is relatively uncommon source of MSCs and therefore differentiation potential should be shown.
Answer: The information was added.
- (b) (c) (d)
Figure 14. The results of hG-MSCs differentiation in three direction: (a) – osteogenic (x100, ARS), (b) – adipogenic (x200, Oil Red O), (c) – hondrogenic (x200, results of histology: immature mesenchymal tissue with a tendency to form cartilage tissue), (d) – hondrogenic (x200, Alcian Blue).
- Flow cytometry plots/histograms from control (stained, non-irradiated cells) should be added to the figures.
Answer: The information was added.
|
(a) |
(b) |
Figure 1. The CD90 expression level one of the donors MSCs 16 days after irradiation at dose of 50 mGy: (а) – preudocolor scaterogamme of stained CD90dim and CD90+ population; (b) – histogram of unstained control (blue), stained non-irradiated control (pink) and CD90dim and CD90+ population of stained irradiated MSCs (black).
|
(a) |
(b) |
Figure 3. The CD105 expression level one of the donors MSCs 16 days after irradiation at dose of 50 mGy: (а) – histogram of unstained control (blue), stained non-irradiated control (pink) and CD105dim and CD105+ population of stained irradiated MSCs (black); (b) – histogram of difference between CD90+/CD105dim and CD90+/CD105+ population (blue).
|
(a) |
(b) |
Figure 4. The CD73 expression level one of the donors MSCs 16 days after irradiation at dose of 50 mGy: (а) – preudocolor scaterogamme of stained CD73dim and CD73+ population; (b) – histogram of unstained control (blue), stained non-irradiated control (pink) and CD73dim and CD73+ population of stained irradiated MSCs (black)..
|
(a) |
(b) |
Figure 6 The CD44 expression level one of the donors MSCs 9 days after irradiation at dose of 50 mGy: (а) – preudocolor scaterogamme of stained CD44dim and CD44+ population; (b) –histogram of unstained control (blue), stained non-irradiated control (pink) and CD44dim and CD44+ population of stained irradiated MSCs (black).
- Mean normalized cell index from all 3 donors should be presented in a graphical way.
Answer: The pictures #11-13 illustrate means of normalized cell index from all 3 donors. There were one experiment for each donor at each time point (n=3).
- The mFISH technique (mentioned once in the method section) is neither described in method section (as separate method) nor any data are presented in results. So it should be either expanded or deleted.
Answer: The mFISH technique was deleted.
- The y axis still do not start from 0 in graphs showing secretion.
Answer: The Y-axis was not started from 0 point for reader convenience and more clearly understanding observed changes.
- As previosly, there are numerous minor points indicated in PDF file comments (attached).
Answer: The article will be corrected before publication in the editorial office of the journal.
Best regards,
Authors.

Reviewer 2 Report
International Journal of Molecular Sciences
COMMENTS TO THE EDITORS AND THE AUTHORS
ijms-2131717R1: “Evaluation of changes in secretory profile and immunophenotype of human mesenchymal stromal cells induced by low doses of ionizing radiation”
Dear the Editor and the Authors,
Please find enclosed the comments for the revised version of the above-mentioned manuscript.
THE OVERALL OPINION OF THE MANUSCRIPT
The authors slightly improved manuscript (the title and the general information related to the statements). However, the major concerns and comments were not addressed: the title is not precisely formulated; the sex-related differences in the responses are not presented; the mechanistic approach is missing; the citation in the introduction and the discussion of the original and the important pioneered results, as well as recent advance in the field focusing on the subject of the study is missing; the intra- and inter- assay coefficients are not presented.
Accordingly, major revision is required for the further consideration.
Please find enclosed some of the suggestions in the comments to the authors listed below.
(1) TITLE
Please consider modifying the title to precisely reflect the results presented in the manuscript. Namely, the results do not show the whole “secretory and immunology profiles”. Please provide only information related to the results presented in the manuscript. The results do not present/show the whole secretory profile nor each possible immunophenotype. Since the results were not obtained using the both sexes, the title cannot state the general knowledge.
(2) ABSTRACT
2.1. Please avoid describing the background in many sentences (1/3 of the text of the abstract describe the background) and please focus on the aim, the methodology and the results of the study.
2.2. Please provide results related to the differences between responses of human MSC obtained from females and males.
2.3. Please provide results related to the mechanistic approach.
(3) INTRODUCTION
3.1. Please describe the original, and the important pioneered results, as well as recent advance in the field focusing on the subject of the study.
(4) MATERIALS AND METHODS
4.1. Please provide the mechanistic studies to prove the physiological significance.
4.2. Please provide the studies related to sex-differences i.e. the studies related to the differences in the responses of human MSC obtained from females and males.
4.3. Please provide intra- as well as inter-assay coefficients for all analyses. Intra-assay coefficients is a measure of the variance between data points within an assay, meaning sample replicates ran within the same plate. Inter-assay coefficients is a measure of the variance between runs of sample replicates on different plates that can be used to assess plate-to-plate consistency.
(5) RESULTS
5.1. Please provide the results obtained from the new experiments related to the mechanistic approach.
5.2. Please provide the results related to the differences in the responses of human MSC obtained from females and males.
(6) DISCUSSION
6.1. Please discuss the original, and important pioneered results, as well as recent advance in the field focusing on the subject of the study.
6.2. Please discuss the new results related to the mechanistic approach.
6.3. Please discuss the results related to the differences in the responses of human MSC obtained from females and males.
(7) REFERENCES
7.1. Please provide references describing the original, and important and pioneered results, but also references describing the recent advance in the field.
(8) FIGURES and FIGURE LEGENDS
8.1. Please provide the new figures and figure legends showing the new results related to the mechanistic approach.
8.2. Please provide the new figures and figure legends showing the new the results related to the differences in the responses of human MSC obtained from females and males.
Good luck and all the best J
Author Response
Dear Reviewer,
Thank you for your attention for details. I have prepared some answers for questions and comments:
THE OVERALL OPINION OF THE MANUSCRIPT
The authors slightly improved manuscript (the title and the general information related to the statements). However, the major concerns and comments were not addressed: the title is not precisely formulated; the sex-related differences in the responses are not presented; the mechanistic approach is missing; the citation in the introduction and the discussion of the original and the important pioneered results, as well as recent advance in the field focusing on the subject of the study is missing; the intra- and inter- assay coefficients are not presented.
(1) TITLE
Please consider modifying the title to precisely reflect the results presented in the manuscript. Namely, the results do not show the whole “secretory and immunology profiles”. Please provide only information related to the results presented in the manuscript. The results do not present/show the whole secretory profile nor each possible immunophenotype. Since the results were not obtained using the both sexes, the title cannot state the general knowledge.
Answer: The following title was suggested: Evaluation of changes in some functional properties of human mesenchymal stromal cells induced by low doses of ionizing radiation.
(2) ABSTRACT
2.1. Please avoid describing the background in many sentences (1/3 of the text of the abstract describe the background) and please focus on the aim, the methodology and the results of the study.
2.2. Please provide results related to the differences between responses of human MSC obtained from females and males.
2.3. Please provide results related to the mechanistic approach.
Answer: The abstract was corrected.
The hG-MSCs cultures were isolated only from three male non-personalized biopsy samples of human connective gingiva tissue, there were not female biopsy samples.
The presented results of MSCs secretory profile and immunophenotype changes under the influence of low radiation doses do not seem sufficient for conducting a more extensive mechanical approach. Today the IRCP still doubts the validity of the threshold or non-threshold concept and we cannot draw serious conclusions based on the data obtained in vitro. However, the observed dose-dependent changes of CD90 expression were analyzed in a similar way. Please, give more information about mechanistic approach if it necessary yet.
(3) INTRODUCTION
3.1. Please describe the original, and the important pioneered results, as well as recent advance in the field focusing on the subject of the study.
Answer: The new information and several links were added to introduction.
“It is important to note that stem cells are characterized by greater radiosensitivity in comparison with other types of cells in the body according to the regularity: the less cell is differentiated, the more cell is radiosensitive. However, it has been shown that the response of MSCs to radiation damage is different from the response of embryonic stem cells. The effect of radiation on embryonic stem cells stimulates them to enter apoptosis [7], whereas the stem cells of adult exhibit a wide range of different options for protection against radiation damage [8]. They are able to compensate for the negative effects of radiation exposure by implementing reactions to the resulting damage, such as the enzymatic activity of ATM protein, activation of cell cycle verification points, repair of double-stranded DNA breaks [9]. Also, the nuclear organization of genetic material of MSCs, which is globally more open and favorable for gene expression, facilitates the process of stopping the cell cycle and DNA repair in damaged cells [10]. In general, it is important to take into account that the degree of radiosensitivity of stem cells is also determined by their age, the stage of the cell cycle [11], the source of production (niche) and the type of radiation affecting them (X-ray, gamma-, beta-, etc.) when conducting research on MSCs models.”
(4) MATERIALS AND METHODS
4.1. Please provide the mechanistic studies to prove the physiological significance.
4.2. Please provide the studies related to sex-differences i.e. the studies related to the differences in the responses of human MSC obtained from females and males.
4.3. Please provide intra- as well as inter-assay coefficients for all analyses. Intra-assay coefficients is a measure of the variance between data points within an assay, meaning sample replicates ran within the same plate. Inter-assay coefficients is a measure of the variance between runs of sample replicates on different plates that can be used to assess plate-to-plate consistency.
Answer: Thanks for your explanation. The information about intra- and inter-assay coefficients was added in 4.6. The enzyme-linked immunosorbent assay (ELISA):
“Intra-assay coefficients of variability (n=25) were less than 10: 9.8 for IL-6, 5.6 for IL-8 and 9.3 for VEGF-A. Inter-assay coefficients of variability (n=2) were less than 15: 8.4 for IL-6, 7.7 for IL-8 and 13.8 for VEGF-A.”
(5) RESULTS
5.1. Please provide the results obtained from the new experiments related to the mechanistic approach.
5.2. Please provide the results related to the differences in the responses of human MSC obtained from females and males.
(6) DISCUSSION
6.1. Please discuss the original, and important pioneered results, as well as recent advance in the field focusing on the subject of the study.
6.2. Please discuss the new results related to the mechanistic approach.
6.3. Please discuss the results related to the differences in the responses of human MSC obtained from females and males.
(7) REFERENCES
7.1. Please provide references describing the original, and important and pioneered results, but also references describing the recent advance in the field.
(8) FIGURES and FIGURE LEGENDS
8.1. Please provide the new figures and figure legends showing the new results related to the mechanistic approach.
8.2. Please provide the new figures and figure legends showing the new the results related to the differences in the responses of human MSC obtained from females and males.
Best regards,
Authors.

Reviewer 3 Report
Thanks to the authors for addressing my comments and editing the figures as suggested.
Round 3
Reviewer 2 Report
International Journal of Molecular Sciences
COMMENTS TO THE EDITORS AND THE AUTHORS
ijms-2131717R2: “Evaluation of changes in some functional properties of human mesenchymal stromal cells induced by low doses of ionizing radiation”
Dear the Editor and the Authors,
Please find enclosed the comments for the revised version of the above-mentioned manuscript.
THE OVERALL OPINION OF THE MANUSCRIPT
The authors slightly improved manuscript (the title – in the responses, but not in the file of the manuscript; the part of the abstract; the introduction) and the general information related to the statements). However, the major concerns and comments were not addressed: the sex-related differences in the responses are not presented; the mechanistic approach is missing.
Accordingly, major revision is required for the further consideration.
Since we are not “moving” forward with the revision and I am the only reviewer, I would greatly appreciate if Editors will appoint at least one more reviewer. Thank you in advance.
Please find enclosed some of the suggestions in the comments to the authors listed below.
(1) ABSTRACT
1.1. Please provide results related to the differences between responses of human MSC obtained from females and males.
1.2. Please provide results related to the mechanistic approach.
(2) MATERIALS AND METHODS
2.1. Please provide the mechanistic studies to prove the physiological significance.
2.2. Please provide the studies related to sex-differences i.e. the studies related to the differences in the responses of human MSC obtained from females and males.
(3) RESULTS
3.1. Please provide the results obtained from the new experiments related to the mechanistic approach.
3.2. Please provide the results related to the differences in the responses of human MSC obtained from females and males.
(4) DISCUSSION
4.1. Please discuss the new results related to the mechanistic approach.
4.2. Please discuss the results related to the differences in the responses of human MSC obtained from females and males.
(5) FIGURES and FIGURE LEGENDS
5.1. Please provide the new figures and figure legends showing the new results related to the mechanistic approach.
5.2. Please provide the new figures and figure legends showing the new the results related to the differences in the responses of human MSC obtained from females and males.
Good luck and all the best J
Author Response
Dear Reviewer,
Thank you for your opinion. I have prepared some answers for questions and comments:
THE OVERALL OPINION OF THE MANUSCRIPT
Answer: The title was corrected in manuscript.
(1) ABSTRACT
1.1. Please provide results related to the differences between responses of human MSC obtained from females and males.
Answer: The results were added to introduction.
“It is also important that the radiosensitivity of MSCs may depend on the gender of the donor. In studies on animal models, it has been repeatedly shown that female individuals are characterized by more pronounced radiosensitivity: changes in the proteome [12], gene expression [13], in particular, oncogenes and proto-oncogenes [14], brain function [15], cognitive abilities [16], etc. Similar patterns are also observed for women and men in epidemiological studies [17, 18].”
1.2. Please provide results related to the mechanistic approach.
Answer: Dear reviewer, please give us more information about the mechanistic approach. My colleagues and I doubt that we understand the term "mechanistic approach" correctly and we ask you to give a detailed explanation for that we can make the necessary adjustments in the manuscript.
Also there is the explanation that we gave earlier:
“The presented results of MSCs secretory profile and immunophenotype changes under the influence of low radiation doses do not seem sufficient for conducting a more extensive mechanical approach. Today the IRCP still doubts the validity of the threshold or non-threshold concept and we cannot draw serious conclusions based on the data obtained in vitro. However, the observed dose-dependent changes of CD90 expression were analyzed in a similar The presented results of MSCs secretory profile and immunophenotype changes under the influence of low radiation doses do not seem sufficient for conducting a more extensive mechanical approach. Today the IRCP still doubts the validity of the threshold or non-threshold concept and we cannot draw serious conclusions based on the data obtained in vitro. However, the observed dose-dependent changes of CD90 expression were analyzed in a similar way.”
(2) MATERIALS AND METHODS
2.1. Please provide the mechanistic studies to prove the physiological significance.
Answer: Dear reviewer, please give us more information about the mechanistic approach. My colleagues and I doubt that we understand the term "mechanistic approach" correctly and we ask you to give a detailed explanation for that we can make the necessary adjustments in the manuscript.
2.2. Please provide the studies related to sex-differences i.e. the studies related to the differences in the responses of human MSC obtained from females and males.
Answer: The hG-MSCs cultures were isolated only from three male non-personalized biopsy samples of human connective gingiva tissue, there were not female biopsy samples.
(3) RESULTS
3.1. Please provide the results obtained from the new experiments related to the mechanistic approach.
Answer: Dear reviewer, please give us more information about the mechanistic approach. My colleagues and I doubt that we understand the term "mechanistic approach" correctly and we ask you to give a detailed explanation for that we can make the necessary adjustments in the manuscript.
3.2. Please provide the results related to the differences in the responses of human MSC obtained from females and males.
Answer: The hG-MSCs cultures were isolated only from three male non-personalized biopsy samples of human connective gingiva tissue, there were not female biopsy samples.
(4) DISCUSSION
4.1. Please discuss the new results related to the mechanistic approach.
Answer: Dear reviewer, please give us more information about the mechanistic approach. My colleagues and I doubt that we understand the term "mechanistic approach" correctly and we ask you to give a detailed explanation for that we can make the necessary adjustments in the manuscript.
4.2. Please discuss the results related to the differences in the responses of human MSC obtained from females and males.
Answer: The hG-MSCs cultures were isolated only from three male non-personalized biopsy samples of human connective gingiva tissue, there were not female biopsy samples.
(5) FIGURES and FIGURE LEGENDS
5.1. Please provide the new figures and figure legends showing the new results related to the mechanistic approach.
Answer: Dear reviewer, please give us more information about the mechanistic approach. My colleagues and I doubt that we understand the term "mechanistic approach" correctly and we ask you to give a detailed explanation for that we can make the necessary adjustments in the manuscript.
5.2. Please provide the new figures and figure legends showing the new the results related to the differences in the responses of human MSC obtained from females and males.
Answer: The hG-MSCs cultures were isolated only from three male non-personalized biopsy samples of human connective gingiva tissue, there were not female biopsy samples.
Best regards,
Authors.

Round 4
Reviewer 2 Report
International Journal of Molecular Sciences
COMMENTS TO THE EDITORS AND THE AUTHORS
ijms-2131717R3: “Evaluation of changes in some functional properties of human mesenchymal stromal cells induced by low doses of ionizing radiation”
Dear the Editor and the Authors,
Please find enclosed the comments for the revised version of the above-mentioned manuscript.
THE OVERALL OPINION OF THE MANUSCRIPT
The authors slightly improved manuscript (the adequate references were included). However, the major concerns and comments were not addressed: the sex-related differences in the responses are not presented; the mechanistic approach is missing.
Accordingly, major revision is required for the further consideration.
As I kindly asked in the previous report, in favor to the authors and the quality of the review process, since we are not “moving” forward with the revision, I would greatly appreciate if Editors will appoint at least one more reviewer. Thank you in advance.
Please find enclosed some of the suggestions in the comments to the authors listed below.
(1) ABSTRACT
1.1. Please provide results related to the differences between responses of human MSC obtained from females and males.
1.2. Please provide results related to the mechanistic approach.
(2) MATERIALS AND METHODS
2.1. Please provide the mechanistic studies to prove the physiological significance.
2.2. Please provide the studies related to sex-differences i.e. the studies related to the differences in the responses of human MSC obtained from females and males.
(3) RESULTS
3.1. Please provide the results obtained from the new experiments related to the mechanistic approach.
3.2. Please provide the results related to the differences in the responses of human MSC obtained from females and males.
(4) DISCUSSION
4.1. Please discuss the new results related to the mechanistic approach.
4.2. Please discuss the results related to the differences in the responses of human MSC obtained from females and males.
(5) FIGURES and FIGURE LEGENDS
5.1. Please provide the new figures and figure legends showing the new results related to the mechanistic approach.
5.2. Please provide the new figures and figure legends showing the new the results related to the differences in the responses of human MSC obtained from females and males.
Good luck and all the best J
Author Response
Dear Reviewer,
Thank you for your opinion. I have prepared some answers for questions and comments:
In the present study, we studied only male MSCs. The purpose of our study was not to compare by gender. We plan to continue research on women's MSCs in the future.
Best regards,
Authors.